# A non-canonical Puf3p-binding sequence regulates *CAT5/COQ7* mRNA under both fermentable and respiratory conditions in budding yeast

**Sachiko Hayashi**  [1]*, **Kazumi Iwamoto**[2], **Tohru Yoshihisa**[1]

**1** Graduate School of Science, University of Hyogo, Ako-gun, Hyogo, Japan, **2** Graduate School of Life Science, University of Hyogo, Ako-gun, Hyogo, Japan

* shayashi@sci.u-hyogo.ac.jp

**Data Availability Statement:** All relevant data are within the paper and its Supporting information files.

**Funding:** This work is supported by JP20K06491 from Japan Society for the Promotion of Science

## Abstract

The *Saccharomyces cerevisiae* uses a highly glycolytic metabolism, if glucose is available, through appropriately suppressing mitochondrial functions except for some of them such as Fe/S cluster biogenesis. Puf3p, a Pumillio family protein, plays a pivotal role in modulating mitochondrial activity, especially during fermentation, by destabilizing its target mRNAs and/ or by repressing their translation. Puf3p preferentially binds to 8-nt conserved binding sequences in the 3′-UTR of nuclear-encoded mitochondrial (nc-mitochondrial) mRNAs, leading to broad effects on gene expression under fermentable conditions. To further explore how Puf3p post-transcriptionally regulates nc-mitochondrial mRNAs in response to cell growth conditions, we initially focused on nc-mitochondrial mRNAs known to be enriched in monosomes in a glucose-rich environment. We unexpectedly found that one of the monosome-enriched mRNAs, *CAT5/COQ7* mRNA, directly interacts with Puf3p through its non-canonical Puf3p binding sequence, which is generally less considered as a Puf3p binding site. Western blot analysis showed that Puf3p represses translation of Cat5p, regardless of culture in fermentable or respiratory medium. *In vitro* binding assay confirmed Puf3p's direct interaction with *CAT5* mRNA *via* this non-canonical Puf3p-binding site. Although *cat5* mutants of the non-canonical Puf3p-binding site grow normally, Cat5p expression is altered, indicating that *CAT5* mRNA is a *bona fide* Puf3p target with additional regulatory factors acting through this sequence. Unlike other yeast PUF proteins, Puf3p uniquely regulates Cat5p by destabilizing mRNA and repressing translation, shedding new light on an unknown part of the Puf3p regulatory network. Given that pathological variants of human *COQ7* lead to $CoQ_{10}$ deficiency and yeast *cat5*Δ can be complemented by *hCOQ7*, our findings may also offer some insights into clinical aspects of *COQ7*-related disorders.

## Introduction

Mitochondria play crucial roles in numerous cellular processes in eukaryotes, including production of the majority of ATP, metabolism of amino acids and lipids, and biosynthesis of various redox molecules such as heme, Fe/S clusters, and coenzymes [1–3]. *Saccharomyces*

(JSPS) KAKENHI (https://www.jsps.go.jp/english/), awarded to SH; 17H05672, 21H05726 and 23K18100 from JSPS KAKENHI, and Specific Research Grants from Takeda Science Foundation (https://www.takeda-sci.or.jp/en/), awarded to TY. The funders had no role in study design, data collection and analysis, decision to publish, or preparation of the manuscript.

**Competing interests:** The authors have declared that no competing interests exist.

*cerevisiae*, a classical model organism for mitochondrial research, normally prefers fermentation rather than respiration. The fermentation process using glucose as substrate is catalytically more efficient than mitochondrial respiration in terms of ATP production per unit protein mass [4], whereas the latter generates 10 times more ATP per glucose molecule [5]. Yeasts only switch to aerobic respiration if glucose is exhausted, which is accompanied by the upregulation of mitochondrial biogenesis. This process is referred to as the diauxic shift. Although yeasts can survive with defects in oxidative phosphorylation coupled with the complete loss of mtDNA, defects in the mitochondrial assembly of Fe/S clusters are lethal [2, 3, 6]. The mitochondrial iron-sulfur cluster assembly machinery is required for the biogenesis of all cellular Fe/S proteins, including the cytosolic and nuclear Fe/S proteins that are involved in DNA maintenance and RNA modification, as well as for cell viability [2, 7].

In *S. cerevisiae*, only eight mitochondrial proteins are encoded by the mitochondrial genome, and all the remaining mitochondrial proteins (>99%) are encoded by the nuclear genome and translated in cytosolic ribosomes in their precursor forms [8–10]. Thus, the correct sorting of mitochondrial proteins is the first step in ensuring organellar functionality. A classical targeting pathway for mitochondrial proteins uses mitochondrial targeting sequences (MTS) that are principally located at their N-terminus [10, 11], whereas approximately one-half of the nuclear mRNAs encoding mitochondrial proteins (nc-mitochondrial mRNAs) are transported to the mitochondrial surface, and translated locally [12–15]. A molecular biological approach using proximity-specific ribosome profiling and a cytological method with electron cryotomography showed that the mitochondrial surface is a place of local translation of nc-mitochondrial mRNAs [16, 17]. Mutant analyses revealed the close relationship between the local translation on the cytosolic surface of mitochondria and mitochondrial functions [18, 19]. Proper translational control of nc-mitochondrial mRNAs is expected in harmony.

Indeed, a member of the Pumilio-homology domain family, Puf3p is a well-known regulator of nc-mitochondrial mRNAs [20–22]. A global analysis showed that Puf3p physically interacts with at least 220 transcripts, of which >70% are nc-mitochondrial mRNAs [23]. Multi-omics analyses have consistently shown binding specificity of Puf3p for nc-mitochondrial mRNAs [24–26]. Puf3p possesses eight-Puf repeats, each of which comprises three α-helices, and the helices of neighboring repeats are stacked to form a crescent shape [27–29]. X-ray crystallography has revealed that three amino acid residues within each Puf repeat directly interact with a single RNA base, and this is the principal determinant of binding specificity [29–34]. The Puf3p repeat domain (Puf3-RD) is sufficient to interacts with target mRNAs and to regulate mRNA metabolism, as illustrated by the binding of yeast Puf3p to the 3′-UTR of *COX17* mRNA [35, 36]. An 8-nt UGUANAUA sequence has been identified as the consensus Puf3p-binding motif on target mRNAs [23, 37, 38]. PUF family proteins generally bind to an 8-nt sequence that includes UGU(A/G) at the 5′ end, along with a more variable 3′ sequence that is specific to the individual PUF proteins [23, 33, 39, 40].

The *puf3Δ* yeast grows slowly in respiratory media [23, 41] and shows impairments in mitochondrial motility and biogenesis [41, 42]. The absence of Puf3p alters the cellular tolerance to oxidative stress and the glutathione redox state [43], as well as increasing cellular oxygen consumption in a growth-dependent manner [44]. At the molecular level, a number of studies have shown that Puf3p destabilizes a wide variety of target mRNAs by promoting deadenylation and reduces the efficiency of their translation, causing a downregulation of mitochondrial biogenesis under fermentable conditions [23, 36, 37, 44–49]. Consistent with a repressive role of Puf3p in glucose-rich medium, Puf3p expression is downregulated during the diauxic shift [42], but it interacts with actively translating polysomes upon glucose depletion and promotes mitochondrial biogenesis [41, 49]. These bidirectional functions of Puf3p are regulated through phosphorylation by the casein kinase Hrr25p [41, 50]. Corresponding *PUF3*(24A)

mutations are dominant negative, and even more strongly inhibit cell growth following the diauxic shift than the complete deletion of *PUF3* [41], which implies that abnormal phosphorylation of Puf3p has a substantial impact on cellular homeostasis.

Yeast *CAT5/COQ7* mRNA encodes a putative monooxygenase required for coenzyme Q (CoQ) biosynthesis and its product Cat5p is an integral membrane protein in the inner mitochondrial membrane [51–53]. Cat5p expression is regulated at the level of mRNA, especially in response to carbon source and CoQ-related metabolites though precise molecular mechanism of this regulation is still under investigation [51, 54]. The functional conservation of Cat5p/Coq7p among species is shown by the ability of human *COQ7* (hCOQ7) to rescue yeast $CoQ_6$ deficiency caused by *cat5Δ* [55, 56]. Indeed, expression/stability of hCOQ7 is fine-tuned by the level of hCOQ4, and is also affected by 2,4-dihydroxybenzoic acid, which is capable of bypassing the enzymatic step performed by hCOQ7 [57, 58]. Such fine-tuning may be achieved at the mRNA level, like the case of yeast Cat5p. Various RNA-binding proteins like HuR and hnRNP C1/C2 interact with the 3′-UTR of hCOQ7 mRNA to modulate hCOQ7 levels, thereby controlling $CoQ_{10}$ [59]. The human Pumilio proteins (PUMs), PUM1 and PUM2, also bind to the 3′-UTR of hCOQ7 mRNA *via* their canonical binding motif, while the expression level of hCOQ7 mRNA is unchanged in the PUM-knockdown cells [60]. Mutations in hCOQ7 are associated with primary ubiquinone deficiency, which contributes to $CoQ_{10}$ deficiency syndrome, and related diseases, predominantly featuring hypertonia and sensorineural hearing loss (SNHL) [57, 61–63]. To fully understand the pathogenesis of $CoQ_{10}$-deficiency related diseases, not only enzymatic mechanism of COQ proteins but also regulation of their expression needs to be clarified.

As described above, a large quantity of information has been accumulated regarding Puf proteins, especially Puf3p in the yeast, from its binding consensus to its modes of action. However, it is still not fully understood how Puf3p utilizes its bidirectional modes of regulation of individual mRNAs to achieve differential regulation of such targets under fermentable and respiratory conditions. The *CAT5/COQ7* mRNA was previously considered not to be a target of Puf3p. However, our *in vivo* and in *vitro* analyses clearly revealed that Puf3p regulates *CAT5* mRNA translation and stability, and directly binds to the *CAT5* 3′-UTR *via* a non-canonical Puf3p-binding sequence (UGUAUAAA) that contains an unusual nucleotide substitution of A for U at position 7. We also found that *CAT5* mRNA expression is regulated by other factors through interactions with this non-canonical Puf3p-binding site at both the transcriptional and post-transcriptional levels. Thus, the present data provide several lines of evidence regarding Puf3p-related regulation of target molecules according to the carbon metabolic state, as well as demonstrating wider recognition of mRNA species by Puf3p. Our finding would contribute to understanding the molecular regulatory mechanisms of the CoQ pathway in higher eukaryotes and give some clues to pathological studies of the $CoQ_{10}$ deficiency in humans.

## Materials and methods

### Yeast strains, plasmids and culture conditions

Standard yeast genetic techniques and other molecular biological techniques were used [64, 65]. The *S. cerevisiae* strains used are listed in S1 Table. The plasmids and primers are summarized in S2 and S3 Tables, respectively. To generate the *cat5Δ puf3Δ* strain (SHSC0090), an amplified *puf3Δ::CgHIS3* fragment was introduced into a *cat5Δ* strain (SHSC0060), and to make the *cat5-101* strain (SHSC0279), a mutant *CAT5* 3′-UTR allele was integrated using the two-step gene replacement strategy. First, the *URA3* marker was integrated at the corresponding transcribed region of the *CAT5* gene, generating the *cat5Δ::URA3* strain (SHSC0268). Subsequently, a 1.08-kb *Eco*RV–*Hind*III fragment containing the *CAT5* 3′-UTR mutant allele

from pSHSC009 was integrated into SHSC0268 to replace the *URA3* marker. 5-Fluoroorotic acid resistant clones were isolated, and correct integration was confirmed by sequencing. To generate a *cat5-101 puf3Δ* strain (SHSC0286), an amplified *puf3Δ::CgHIS3* fragment was introduced into SHSC0279. To make the *cat5-102* (SHSC0480), *cat5-103* (SHSC0484), and *cat5-104* (SHSC0488) strains, which included a substitution at position 7 of the Puf3p-binding sequence in the *CAT5* 3′-UTR, the two-step gene replacement strategy was performed with some modifications: for the *cat5-102*, *cat5-103*, and *cat5-104* strains, 1.08-kb *Eco*RV–*Hind*III fragments containing the *CAT5* 3′-UTR mutant alleles from pSHSC039, pSHSC040, and pSHSC041, respectively, were integrated into SHSC0268 to replace the *URA3* marker.

The yeast strains were cultured at 30˚C in YPD [1.0% (w/v) yeast extract, 2.0% (w/v) polypeptone, and 2.0% (w/v) D-glucose], YPGal [1.0% (w/v) yeast extract, 2.0% (w/v) polypeptone, and 2.0% (w/v) D-galactose], or YPGly [1.0% (w/v) yeast extract, 2.0% (w/v) polypeptone, and 2.0% (w/v) glycerol]. For growth comparisons, cells were cultured in SCD [0.67% (w/v) yeast nitrogen base without amino acids, 0.5% (w/v) casamino acids, vitamin assay, and 2.0% (w/v) D-glucose] or SCGly [0.67% (w/v) yeast nitrogen base without amino acids, 0.5% (w/v) casamino acids, vitamin assay, and 2.0% (w/v) glycerol] with 20 μg/ml appropriate amino acids and nucleobase supplements.

## Crude RNA preparation and northern blotting

Crude RNA was extracted from mid log-phase cultured yeast cells at 65˚C using Na-acetate/sodium dodecyl sulfate (SDS) buffer [50 mM Na-acetate, pH 5.2, 10 mM EDTA, 1.0% (w/v) SDS] and an equal volume of Acidic Phenol Chloroform [phenol:chloroform = 5:1, pH 4.5] or Phenol, Saturated with Citrate Buffer [pH 4.5] (Nacalai Tesque, Kyoto, Japan). The aqueous phase was separated by the addition of chloroform and subsequent centrifugation, and was sequentially extracted using phenol/chloroform and chloroform. The resulting RNA was precipitated using 2-propanol, and the final pellets were dissolved in TE [10 mM Tris-HCl, pH 7.5 and 1.0 mM EDTA]. The crude RNA and the RNAs obtained from sucrose gradient fractions were separated on a 1.2% (w/v) agarose gel, with 2.2 M formaldehyde in the MOPS buffer, and transferred onto Hybond-N$^+$ charged nylon membranes (GE Healthcare, Chicago, IL, USA) by capillary transfer in 20×SSC. Hybridization with digoxigenin (DIG)-labeled antisense RNA probes was performed in DIG Easy Hyb (Roche Diagnostics) at 68˚C. The antisense RNA probes for *AIM17*, *MRPL16*, *RSM10*, and *CAT5/COQ7* were labeled with digoxigenin using DIG Northern Starter Kit (Roche Diagnostics).

## Total protein extraction and western blotting

Mid log-phase yeast cells (0.5 OD$_{660}$ units) were collected by centrifugation, resuspended in 113 μl Lysis Buffer II [10 mM Tris-HCl, 1 mM EDTA-Na, 278 mM NaOH, and 6.2% (v/v) β-mercaptoethanol], and incubated for 5 min on ice. The cell lysates were treated with 1.0 ml of ice-cold 10% (w/v) TCA for 10 min on ice, then centrifuged at $18,000 \times g$ for 5 min at 4˚C in a microcentrifuge. After washing with ice-cold acetone, the final pellets were resuspended in SDS-PAGE Sample Buffer [50 mM Tris-HCl, pH 6.8, 5.0 mM EDTA-Na, pH 8.0, 2.5% (w/v) SDS, 12.5% (w/v) glycerol, 0.005% (w/v) bromophenol blue, 2.0% (v/v) β-mercaptoethanol, and 2.0 mM PMSF] supplemented with Tris to a final concentration of 20 mM, and heated for 5 min at 95˚C. Immunoblots were developed using ECL, with horseradish peroxidase-conjugated goat anti-rabbit IgG as the secondary antibody. For ECL detection, membranes were incubated for 1 min in ECL solution [100 mM Tris-HCl, pH 8.6, 0.2 mM *p*-coumaric acid, 1.2 mM luminol sodium salt, and 0.01% (w/v) H$_2$O$_2$] [66]. Antibodies against Mrpl16p and

Cat5p/Coq7p were kindly provided by Prof. Antonio Barrientos (University of Miami, USA) and Prof. Catherine F. Clarke (UCLA, USA), respectively.

## Protein expression and purification

The GST-Puf3RD expression plasmid, pTYE600, and its vector, pGEX-4T-2, were introduced into the *Escherichia coli* strains BL-21 (DE3) or TG1 [*supE hsd*Δ5, *thi*, Δ(*lac-proAB*)/F', *traD*36, *proAB*$^+$, *lacI*$^q$, *lacZ*ΔM15]. GST-tagged fusion proteins were overexpressed by incubating cultures to log-phase growth with 0.2 mM IPTG at 37°C for 2 hr. The collected cells were washed with ice-cold STE Buffer [10 mM Tris-HCl, pH 8.0, 0.1 M NaCl, and 1 mM EDTA-Na], and resuspended in ice-cold Lysis Buffer III [50 mM Tris-HCl, pH 7.5, 100 mM NaCl, 2 mM EDTA-Na, and 1 mM PMSF]. The cells were then disrupted using an ultrasonic disruptor (UD-201; Tomy, Tokyo, Japan). In advance of centrifugation for the removal of cells debris, cell lysates were incubated with 0.1% (w/v) Triton X-100 on ice for 5 min. The GST fusion proteins were purified using Glutathione Sepharose™ 4B (GE Healthcare), Binding Buffer I [50 mM Tris-HCl, pH 7.5, 0.1% (w/v) Triton X-100, 2 mM EDTA-Na, pH 8.0, 0.1 M NaCl, and 1 mM PMSF], and Elution Buffer [50 mM Tris-HCl, pH 8.0 and 25 mM reduced glutathione], according to the manufacturer's protocol. Protein eluates were dialyzed against 50 mM Tris-HCl, pH 8.0, and the protein concentrations of the final samples were determined using the Bradford method with Protein Assay CBB Solution (Nacalai Tesque) and bovine serum albumin (Fujifilm Wako Pure Chemical) as a standard.

## *In vitro* transcription and fluorescence labeling of RNA

DNA templates were amplified from pTYE611 (*MRPL16* 3′-UTR), pTYE612 [*mrpl16* 3′-UTR w/o Puf3 site (26TGTA to 26ACAC)], pSHE002 [*mrpl16* 3′-UTR double mutations (18TGTA to 18ACAC, 26TGTA to 26ACAC)], pTYE613 (*CAT5/COQ7* 3′-UTR), or pSHE001 [*cat5/coq7* 3′-UTR (92TGTA to 92ACAC)] with MRPL16_3UTR_rv2 or CAT5_3UTR_rv2, and M13-20 as primers. The resulting PCR products were purified using Illustra GFX PCR DNA & Gel Band Purification Kit (GE Healthcare), followed by 2-propanol precipitation, and the final pellets were dissolved in DEPC-treated water (DEPC-DW). *In vitro* transcription was performed using SP6 RNA polymerase with a MEGAscript kit (Ambion, Austin, TX, USA). The transcribed RNAs were subjected to phenol/chloroform extraction, followed by chloroform extraction and 2-propanol precipitation, and the final pellets were dissolved in 40 μl of TE, and were then desalted with NucAway™ Spin Columns (Ambion) equilibrated with TE. The RNAs was treated with NaIO$_4$ at final concentration of 1.8 mg/ml at 23°C for 60 min in the dark, resulting in the oxidization of the 2', 3'-diol at the 3′-terminus of the RNAs to a 2', 3'-dialdehyde. The buffer of the NaIO$_4$-oxidized RNAs was exchanged for 0.10 M NaOAc, pH 5.2 using PD Spin Traps G-25 (GE Healthcare). The recovered RNA samples were each mixed with 7.0 μl of 10 mM Cy3 hydrazide (BroadPharm, San Diego, CA, USA) and incubated at 4°C for 4 hr in the dark to facilitate dialdehyde–hydrazide conjugation. After ethanol precipitation, the final pellets were dissolved in 10 μl of DEPC-DW, and the RNA was purified by PAGE using a 7% (w/v) TBE-urea polyacrylamide gel. The RNA samples were each eluted with 750 μl of Urea-PAGE Elution Buffer (0.30 M NaOAc, pH 5.2, 5.0 mM EDTA-Na, and 0.10% (w/v) SDS) in the dark overnight, and then this process was repeated with 250 μl of the same buffer for 4 hr. The eluted RNA samples were extracted with phenol/chloroform and then with chloroform. After precipitation with 2-propanol in the presence of 12.5 μg/ml glycogen, the final pellets were dissolved in 20 μl DEPC-DW. For the quantification of fluorescence-labeled RNAs, the RNA samples were separated on a 7% (w/v) TBE-urea polyacrylamide gel, and the signals were

detected using a laser scanner (Typhoon FLA-7000; GE Healthcare) or a cooled CCD camera system (Ez-Capture; ATTO, Tokyo, Japan).

### Electrophoretic mobility shift assay (EMSA)

*In vitro* binding analysis of GST-Puf3-RD was performed largely as previously described [35, 36]. Reaction mixtures were prepared in 20 µl of Binding Buffer II [10 mM HEPES-KOH, pH 7.4, 50 mM KCl, 1.0 mM EDTA-Na, 2.0 mM DTT, 200 U/ml RNasin, 0.1 mg/ml bovine serum albumin, 0.01% (w/v) Tween-20, 0.1 mg/ml poly(rU), and 10 µg/ml yeast tRNA] in the presence (0.65 µM) or absence of GST-Puf3-RD or GST, and final 600 pM and 100 pM concentrations of fluorescence-labeled *in vitro* transcripts of the *MRPL16* or *CAT5* 3'-UTRs, respectively. The mixtures were incubated at 24°C for 30 min, 5.0 µg of heparin was added, and they were then re-incubated at 24°C for 10 min. For electrophoresis on a 7% (w/v) TBE-acrylamide gel, the reaction mixtures were mixed with 4.0 µl of 5×Gel-shift Sample Buffer [5×TBE, 25% (w/v) sucrose]. Electrophoresis was performed at 200 V and 4°C for 2–2.5 hr, then fluorescence signals were detected using Typhoon FLA-7000 (GE Healthcare), and the data were processed using Image Quant TL software (GE Healthcare).

### Measurement of protein synthesis

HPG labeling to measure protein synthesis was performed as previously described [67]. Cells grown in SD [0.67% (w/v) yeast nitrogen base without amino acids, and 2.0% (w/v) D-glucose] with appropriate supplements until log phase were transferred to SD with appropriate supplement except for methionine, were incubated for 30 min for methionine starvation, and then total 3.0 $OD_{660}$ cells were labeled with 2.8 nmol of HPG (Jena Bioscience, Jena, Germany) for 30 min. SDS cell lysates were prepared as described above but final pellets were resuspended in 200 µL of HEPES SDS Buffer [100 mM HEPES-NaOH, pH 7.5, 2.0% (w/v) SDS, and 2 mM PMSF]. Click chemistry to conjugate HPG residues with biotin was performed as described [68]. Briefly, the SDS lysates were first treated with final 44 mM sodium ascorbate and 111 mM iodoacetamide for 10 min at room temperature, and they were mixed with 0.10 mM azide-$PEG_3$-biotin (Sigma Aldrich, St. Luis, Missouri, USA), 25 mM $CuSO_4$, 10 mM tris (3-hydroxypropyltriazolylmethyl)amine (Sigma Aldrich), and 20 mM aminoguanidine in final concentrations in this order. After 15 min-incubation at 30°C, unreacted reagents were removed using methanol-chloroform precipitation, and the pelleted proteins were dissolved in 500 µL of IP SDS Buffer [20 mM Tris-HCl, pH 7.5, 150 mM, 1.0% (w/v) SDS, and 2.0 mM PMSF]. After 2-fold dilution with IP Buffer w/o SDS [20 mM Tris-HCl, pH 7.5, 150 mM, and 1.0% (w/v) Triton X-100], biotinylated proteins were recovered with avidin D-agarose (Vector Laboratories, Newark, CA, USA). After extensive wash of the beads with IP Buffer [20 mM Tris-HCl, pH 7.5, 150 mM, 1.0% (w/v) Triton X-100, and 0.10% (w/v) SDS], bound proteins were released from the beads with SDS-PAGE Sample Buffer, and were subjected to western blotting.

### Thiolutin chase

The *cat5-101* and *puf3Δ* mutant strains (SHSC0279 and SHSC0056, respectively), together with the wild-type BY4741 strain, were cultured in YPD to the log phase, and a final concentration of 3.0 µg/ml of thiolutin (Tocris Bioscience, Bristol, United Kingdom) was added. Two-milliliter yeast samples were withdrawn from the cultures at 0, 5, 10, 20, 40, and 60 min after the addition of thiolutin, briefly centrifuged, and frozen in liquid $N_2$. RNA samples were prepared from the yeast samples, and 5.0-µg RNA aliquots were analyzed by northern blotting using an RNA probe against *CAT5* ORF labeled with DIG, as described above.

## Results

### Puf3p functions as a translational repressor of *CAT5* mRNA

We were originally interested in unique translational characteristics of monosome-enriched mRNAs in *S. cerevisiae* reported by Heyer and Moore [69]. A considerable part of monosome-enriched mRNAs was nc-mitochondrial mRNAs, and many of them have the canonical Puf3p-binding site in their 3'-UTR (Table 1).

According to [69], nc-mitochondrial mRNAs shown in the table were categorized into monosome-enriched mRNAs (upper part) or polysome-enriched mRNAs (lower part). Mitochondria-localized mRNAs that are dependent on and independent of Puf3p are highlighted by orange and light yellow, respectively. *Saccharomyces Genome Database* (https://www.yeastgenome.org/) was referred for cellular function and mRNA localization. Saint-Georges *et al.* [70] were referred for Puf3p binding site and MLR (mitochondrial localization of nuclear-encoded mRNAs) classification.

A considerable portion of these mRNAs were also known to be localized on the mitochondrial surface in a Puf3p-dependent manner [70]. During analyses of the individual nc-mitochondrial mRNAs enriched in the monosome fraction, we realized that *CAT5/COQ7* mRNA, which was previously reported to lack the Puf3p-binding site, seems to be akin to the canonical Puf3p target, like *MRPL16* or *RSM10* mRNAs. Northern blotting of several monosome-enriched mRNAs revealed that *CAT5* mRNA modestly but reproducibly increased in a *puf3Δ* mutant and that this difference was similar to those of *MRPL16* and *RSM10* mRNAs (S1 Fig). Such increase in the *puf3Δ* mutant was only seen under the fermentable conditions, and no statistically meaningful increase of mRNAs was observed under respiratory conditions among *CAT5* mRNA and other mRNAs tested here.

We then subsequently verified the effects of *PUF3* deletion on *CAT5* expression at the protein levels. As shown in Fig 1A–1C, the steady-state Mrpl16p expression (Puf3p target control) in the wild-type yeast was very low in the fermentable medium (YPD), but was approximately 8–9 times higher in the respiratory media (YPGal or YPGly; Fig 1B). Similarly, Cat5p expression was substantially induced under these growth conditions, as for that of Mrpl16p (Fig 1D–1F). The wild-type cells expressed approximately 3–4 times more Cat5p in the respiratory media than in the fermentable medium. The effects of *puf3Δ* mutation are more obvious on the protein level than the mRNA level. Importantly, the deletion of Puf3p increased the expression of both Mrpl16p and Cat5p in yeast grown in the fermentable medium, but this effect was less marked when the yeast was grown in the respiratory media (Fig 1C and 1F). In the respiratory media, Puf3p functioned as a translational repressor of Cat5p, but it did not reduce the translation of Mrpl16p. These findings imply that Puf3p acts as a negative regulator of both mRNAs at the post-transcriptional level in a carbon source- and/or transcript-dependent manner.

### Puf3p binds to non-canonical sequences in the *MRPL16* and *CAT5* mRNAs with a variation at position 7 of the Puf3p-binding motif *in vitro*

As shown in Fig 2A, three amino acid residues within each Puf repeat, make direct contact with a single RNA base, which is the principal determinant of binding specificity [29–34]. *CAT5* mRNA does not contain the canonical Puf3p-binding sequence UGUANAUA, but it does contain a similar sequence, UGUAUA<u>A</u>A, which differs at nucleotide 7 of the Puf3p-binding sequence (A instead of U), at position 92–99 nt of the *CAT5* 3'-UTR (Fig 2C). To directly evaluate whether *CAT5* mRNA is a *bona fide* target of Puf3p, we employed an *in vitro* assay of the binding of Puf3p [35, 36] to the experimentally defined 3'-UTRs of the *MRPL16*

**Table 1. Monosome-enriched mRNAs for nc-mitochondrial proteins are inclined to be localized on the mitochondrial membrane *via* Puf3p.**

| Systematic name | Standard name | Cellular function | mRNA localization on mitochondrial outer membarane | Puf3p binding site on the 3'-UTR | MLR Classification | mRNA type |
|---|---|---|---|---|---|---|
| YDR336W | MRX8 | Protein associates with mitochondrial ribosome | Localized | No | independent of Puf3p | Monosome |
| YMR064W | AEP1 | Protein required for expression of the mitochondrial *OLI1* gene | Localized | Yes | depends on Puf3p | Monosome |
| YDR332W | IRC3 | Double-stranded DNA-dependent helicase | undetermined | No | - | Monosome |
| YER024W | YAT2 | Carnitine acetyltransferase | Localized | No | independent of Puf3p | Monosome |
| YGR015C | YGR015C | Mitochondria-localized protein | undetermined | No | - | Monosome |
| YJL023C | PET130 | Protein required for respiratory growth | Localized | No | independent of Puf3p | Monosome |
| YML042W | CAT2 | Carnitine acetyl-CoA transferase | Localized | No | independent of Puf3p | Monosome |
| YBL107C | MIX23 | Mitochondrial intermembrane space protein | undetermined | - | - | Monosome |
| YER015W | FAA2 | Medium chain fatty acyl-CoA synthetase | undetermined | - | - | Monosome |
| YLR105C | SEN2 | Subunit of the tRNA splicing endonuclease | undetermined | - | - | Monosome |
| YBL038W | MRPL16 | Mitochondrial ribosomal protein of the large subunit | Localized | Yes | depends on Puf3p | Monosome |
| YCR005C | CIT2 | Citrate synthase | Localized | No | independent of Puf3p | Monosome |
| YCR024C | SLM5 | Mitochondrial asparaginyl-tRNA synthetase | Localized | Yes | depends on Puf3p | Monosome |
| YDL044C | MTF2 | Protein interacts with mitochondrial RNA polymerase | Localized | Yes | depends on Puf3p | Monosome |
| YDR041W | RSM10 | Mitochondrial ribosomal protein of the small subunit | Localized | Yes | depends on Puf3p | Monosome |
| YDR191W | HST4 | NAD(+)-dependent protein deacetylase | undetermined | - | - | Monosome |
| YDR197W | CBS2 | Mitochondrial translational activator of the *COB* mRNA | Localized | Yes | depends on Puf3p | Monosome |
| YER183C | FAU1 | 5,10-methenyltetrahydrofolate synthetase | Not localized | No | - | Monosome |
| YGR146C | ECL1 | mitochondrial-dependent role in the extension of chronological lifespan | undetermined | - | - | Monosome |
| YHL021C | AIM17 | Mitochondria-localized protein | Localized | No | independent of Puf3p | Monosome |
| YJL133W | MRS3 | Iron transporter | Localized | No | independent of Puf3p | Monosome |
| YJL180C | ATP12 | Assembly factor for F1 sector of mitochondrial F1F0 ATP synthase | Localized | Yes | depends on Puf3p | Monosome |
| YJL209W | CBP1 | Mitochondrial protein, regulator of *COB* mRNA stability and translation | Localized | Yes | depends on Puf3p | Monosome |
| YNL213C | RRG9 | Mitochondria-localized protein | Localized | No | independent of Puf3p | Monosome |
| YOR037W | CYC2 | Mitochondrial peripheral inner membrane protein | Localized | No | independent of Puf3p | Monosome |
| YOR125C | CAT5 | Protein required for Coenzyme Q biosynthesis | Localized | No | independent of Puf3p | Monosome |
| YLR355C | ILV5 | Acetohydroxyacid reductoisomerase and mtDNA binding protein | Not localized | No | - | Polysome |
| YGR094W | VAS1 | Mitochondrial and cytoplasmic valyl-tRNA synthetase | Localized | No | independent of Puf3p | Polysome |
| YOR335C | ALA1 | Cytoplasmic and mitochondrial alanyl-tRNA synthetase | undetermined | Yes | - | Polysome |

(*Continued*)

**Table 1.** (Continued)

| Systematic name | Standard name | Cellular function | mRNA localization on mitochondrial outer membarane | Puf3p binding site on the 3'-UTR | MLR Classification | mRNA type |
|---|---|---|---|---|---|---|
| YPL061W | *ALD6* | Cytosolic aldehyde dehydrogenase | undetermined | No | - | Polysome |
| YIL125W | *KGD1* | Subunit of the mitochondrial alpha-ketoglutarate dehydrogenase complex | Localized | No | independent of Puf3p | Polysome |
| YKL182W | *FAS1* | Beta subunit of fatty acid synthetase | Not localized | No | - | Polysome |
| YPL231W | *FAS2* | Alpha subunit of fatty acid synthetase | undetermined | No | - | Polysome |
| YNR016C | *ACC1* | Acetyl-CoA carboxylase, biotin containing enzyme | undetermined | No | - | Polysome |

and *CAT5* mRNAs [71]. We purified the glutathione *S*-transferase-tagged Puf3p repeat domain (Puf3-RD; 465–879 amino acids) expressed in *E. coli*, and *in vitro*-transcribed the 3'-UTRs of the Cy3-labeled *MRPL16* and *CAT5* mRNAs with or without mutations. These materials were subjected to electrophoretic mobility shift assay (EMSA). Various concentrations of

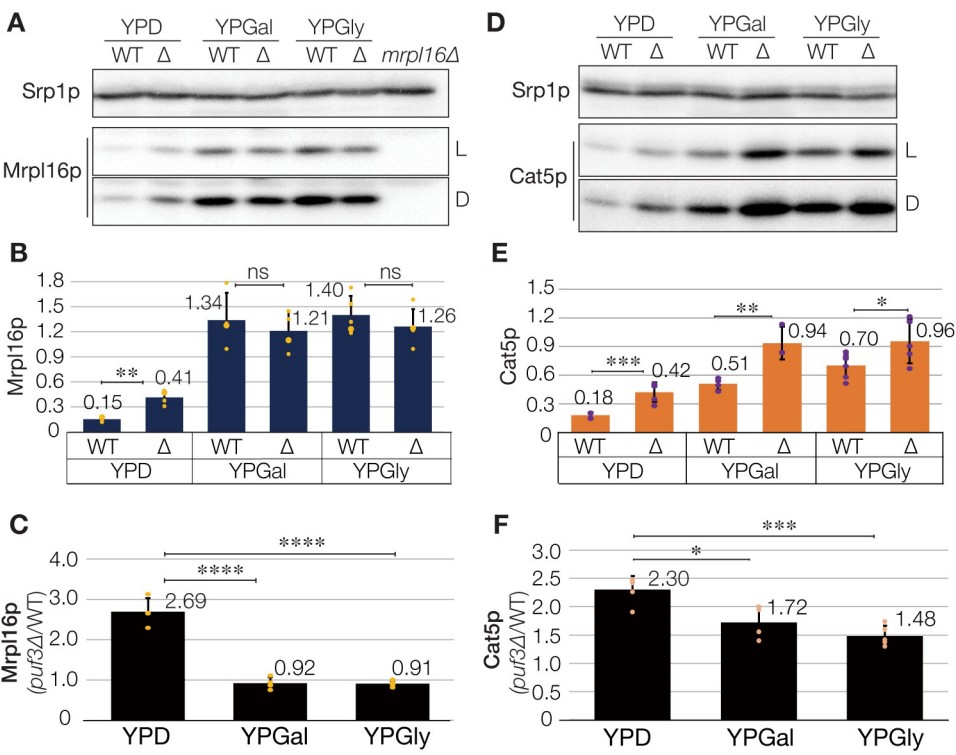

**Fig 1. Puf3p regulates the expression of Mrpl16p and Cat5p in a carbon source-dependent manner.** (A) Steady-state levels of Mrpl16p in the wild-type (WT) and *puf3Δ* strains. Total cell extracts prepared from the same mass of cells were analyzed by immunoblotting using antibodies specific for Mrpl16p, and for Srp1p as a loading control. Because the expression levels of Mrpl16p differed significantly, light-contrast (L) and dark-contrast (D) images of the same immunoblot are displayed. (B) Bar chart showing the relative abundance of Mrpl16p, normalized with that of Srp1p, for more than three biological replicates in (A) (Student's *t*-test, **, $p < 0.01$; ns, not significant). (C) Relative expression of Mrpl16p in the *puf3Δ* cells *versus* that in wild-type cells under various culture conditions. The mean value for the WT cells in YPD was set to 1.0 (n ≥ 4; ****, $p < 0.0001$). (D) Steady-state levels of Cat5p in the WT and *puf3Δ* strains. Cell extracts were analyzed as in (A). (E) Bar chart showing the relative abundance of Cat5p, quantified from at least three biological replicates in (D), and shown as in (B) (Student's *t*-test, ***, $p < 0.001$; **, $p < 0.01$; *, $p < 0.05$). (F) Relative expression of Cat5p in the *puf3Δ* cells *versus* that in the WT cells (n ≥ 4; *, $p < 0.05$; ***, $p < 0.001$).

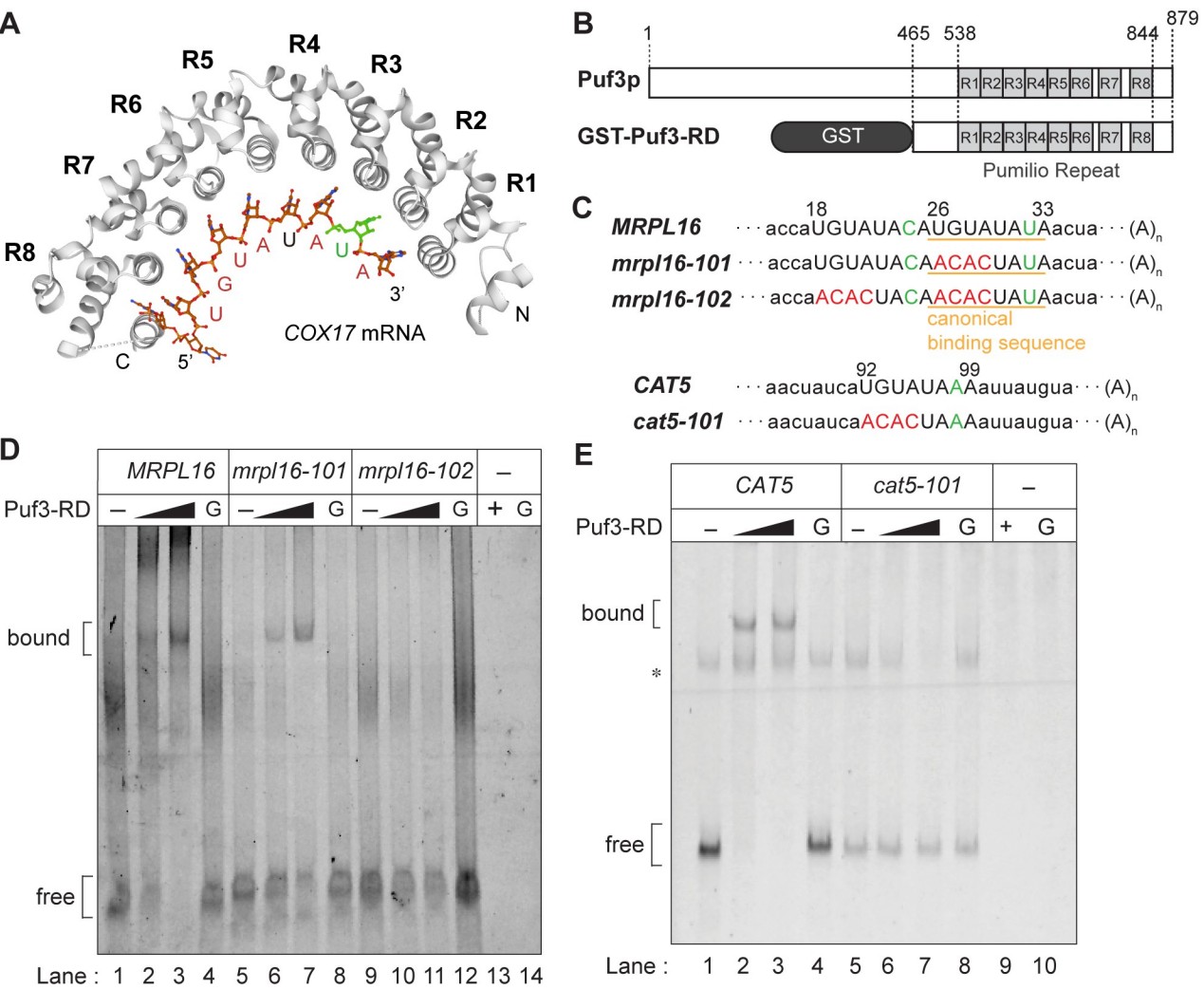

**Fig 2. Puf3p accepts variants at nucleotide 7 of the Puf3p canonical binding sequence in the *MRPL16* and *CAT5* 3'-UTRs *in vitro*.** (A) Crystal structure of the Puf3p repeat domain (Puf3-RD) in complex with the 3'-UTR of the *COX17* mRNA (PDB ID code: 3K4E). R1–R8 indicate the PUF repeats of Puf3-RD. The nucleotides of *COX17* mRNA (5'-UGUAUAUA-3') are shown in green for position 7 of the canonical Puf3p-binding sequence and in red for the other positions. (B) Overview of the Puf3-RD fusion protein used in this study. The numbers represent the amino acid positions in Puf3p. (C) The upper scheme shows the 3'-UTR sequences of the *MRPL16* mRNA used in EMSA. The numbers represent the nucleotide positions in the 3'-UTR. Capital letters indicate the canonical Puf3p-binding sequence (orange underline; U26–A33) or a candidate for a non-canonical Puf3p-binding sequence with a different nucleotide at position 7 (24C; U18–A25). Nucleotide 7 of the canonical or non-canonical Puf3p-binding sequences is shown in green and mutations of the canonical or non-canonical Puf3p-binding sequences are shown in red. The lower scheme shows the 3'-UTR sequences of the *CAT5* mRNA used in EMSA. The numbers represent the nucleotide positions in the 3'-UTR. Capital letters represent a candidate non-canonical Puf3p-binding sequence (U92–A99), which has a different nucleotide at position 7 (98A; green). The mutated positions are highlighted in red. (D) EMSA was performed using the wild-type and mutant forms of Cy3-labeled RNAs corresponding to the *MRPL16* mRNA 3'-UTRs; 600 pmol of the Cy3-labeled RNAs were used as substrates. The final concentrations of GST-Puf3-RD used in the assays were 0 µM (–), 0.65 µM, and 1.95 µM (black triangle). Lane G (lanes 4, 8, and 12) is the control gel shift with 1.95 µM GST. No RNA substrates were included in lanes 13 or 14, in the presence of 1.95 µM GST-Puf3p-RD (+) or GST (G), respectively. (E) EMSA performed using the wild-type and mutant forms of the *CAT5* 3'-UTR. The assay conditions were similar to those described in (D), except 100 pmol Cy3-labeled RNAs were included instead of 600 pmol. Bands marked by an asterisk may represent RNA dimers.

purified recombinant Puf3-RD (0–1.95 µM) were incubated with fixed amounts of labeled wild-type or mutated RNAs (Fig 2D and 2E).

As shown in Fig 2D, Puf3p dose-dependently bound to the wild-type *MRPL16* 3'-UTR, gradually yielding a shifted band as increasing amount of the GST-Puf3-RD fusion were used

in the EMSA (lanes 1–3). The Puf3-RD part was responsible for this binding, because there was no band shift in the presence of GST alone (G; lane 4). Strikingly, *mrpl16-101*, in which the first four nucleotides of the canonical sequence UGUA were replaced by ACAC, still showed a clear band shift that depended on the amount of Puf3-RD present (Fig 2D, lanes 5–7). According to Jackson *et al.* [36], the substitution of the UGUA sequence in Puf3p targets is sufficient to abolish their binding affinity with Puf3p. Nevertheless, the *mrpl16-101* 3′-UTR with this substitution retained its binding affinity with Puf3-RD. We then searched for an additional Puf3p-binding sequence within the 64 nt of the *MRPL16* 3′-UTR, and found one sequence that was similar to the Puf3p-binding sequence, UGUAUA<u>C</u>A, an 8-nt with a U-to-C substitution at position 7, immediately upstream of the canonical Puf3p-binding sequence. We then generated an *mrpl16-102* mutant with additional substitutions of UGUA for ACAC in a candidate Puf3p-binding sequence in *mrpl16-101*, and then performed EMSA (Fig 2D). The *mrpl16-102* 3′-UTR showed no band shift, irrespective of the presence or absence of Puf3-RD, and all the mutant RNAs remained as the lower free RNA bands (Fig 2D, lanes 9–11). These *in vitro* results indicated that the *MRPL16* mRNA contains two distinct Puf3p-binding sites, the canonical one and a variant, which has a single-nucleotide substitution at position 7 of the canonical sequence.

Next, we prepared the 115-nt sequence of the *CAT5* mRNA 3′-UTR as a wild-type substrate and performed EMSA, as shown in Fig 2E. This showed that the wild-type *CAT5* 3′-UTR directly interacts with Puf3-RD (Fig 2E, lanes 1–4). The shifted band appeared in the presence of Puf3-RD, but not in its absence or in the presence of GST alone. Conversely, the lowest free RNA band disappeared when Puf3-RD was incubated with the wild-type *CAT5* 3′-UTR, likewise *MRPL16* (Fig 2E, lanes 2 and 3). Next, we tested whether a mutated version of the *CAT5* mRNA, *cat5-101*, which includes the UGUA-to-ACAC mutation in the candidate non-canonical Puf3p-binding sequence (Fig 2C), binds to Puf3-RD *in vitro*. As expected, the *cat5-101* 3′-UTR did not bind to Puf3-RD (Fig 2E, lanes 5–7). Therefore, we concluded that the *CAT5* mRNA is a novel *bona fide* Puf3p target that interacts with Puf3p *via* the non-canonical Puf3p-binding sequence UGUAUAAA *in vitro*.

## The non-canonical Puf3p-binding sequence of the *CAT5* mRNA is required for the appropriate expression of Cat5p under both fermentable and respiratory conditions

To clarify whether the novel Puf3p-binding sequence of *CAT5* mRNA identified in the *in vitro* analyses is functional *in vivo*, we generated a strain containing the UGUA-to-ACAC mutation in the non-canonical Puf3p-binding sequence of the *CAT5* gene (*cat5-101*) described above. As shown in Fig 3A, the *cat5-101* strain grew similarly to the wild-type cells, both in the fermentable and the respiratory (YPD and YPGly) media at 30˚C and 37˚C (S2A Fig). By contrast, the *cat5*-deleted strains (*cat5Δ* and *cat5Δ puf3Δ*) grew on YPD but not on YPGly at 30˚C or at 37˚C (Fig 3A and S2A Fig), consistent with their previously reported respiration-deficient phenotypes [53]. Because Puf3p seems to regulate Cat5p expression post-transcriptionally (Fig 1D–1F), the Puf3p-binding affinity for the *CAT5* mRNA may affect the translation of Cat5p. Unexpected as a repressive function of Puf3p, the *cat5-101* strain produced less Cat5p than the wild-type strain in YPD medium (Fig 3B, lanes 1 and 2; Student's *t*-test, $p < 0.001$). Importantly, the amount of Cat5p produced in the *cat5-101* mutant was similar, regardless of the presence or absence of Puf3p (Fig 3B, relative amount (RA) = 0.33±0.11 and 0.43±0.15, respectively; Student's *t*-test, $p = 0.209$). Given that the *puf3Δ* strain showed higher expression of wild-type Cat5p (Fig 3B, lane 4, RA = 2.25±0.87, Student's *t*-test, $p = 0.034$; Fig 1D–1F), the low Cat5p expression in the *cat5-101* mutant may be the result of another protein binding to

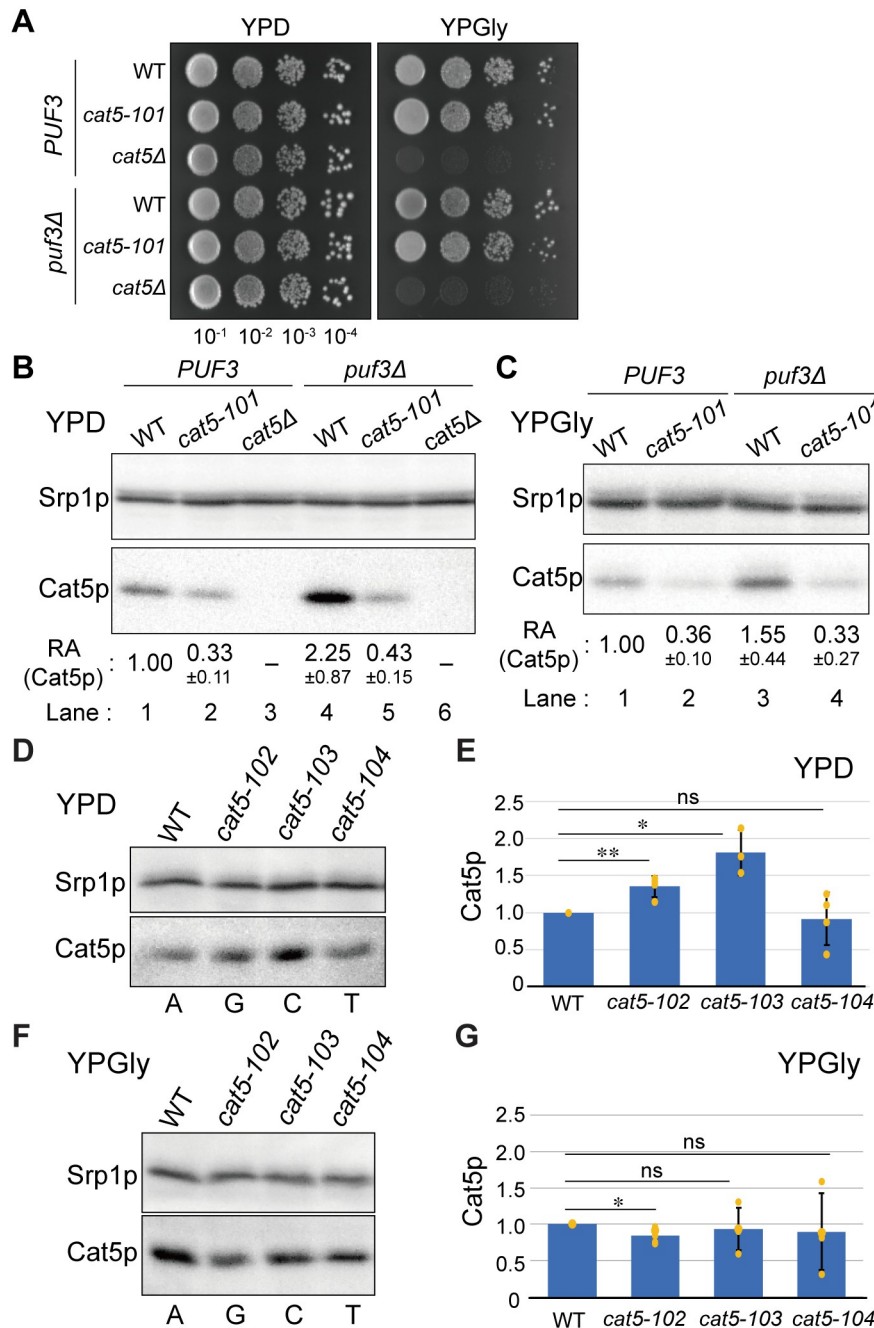

**Fig 3. Mutations in the non-canonical Puf3p-binding sequence of *CAT5* mRNA affect Cat5p expression *in vivo* but not cell viability.** (A) Comparison of the growth of the wild-type (WT), *cat5-101*, and *cat5Δ* strains in the presence or absence of the *PUF3* gene. Saturated cultures of the indicated strains were serially diluted 10-fold, dropped onto YPD or YPGly plates, and cultured at 30˚C. (B)–(C) Western blot analysis of Cat5p expression in the corresponding yeast strains. Cells were cultured at 30˚C in YPD (B) or in YPGly (C) media. Srp1p was used as a loading control. The numbers under the gel images represent the mean ± standard deviation of the relative amount (RA) of Cat5p, quantified from n ≥ 3. The WT Cat5p expression was set to 1.00. (D)–(G) Quantitative western blot analysis of Cat5p expression in the WT strain and in strains with one of the three point mutants of the *CAT5* Puf3p-binding sequence at position 7 (*cat5-102*, A-to-G; *cat5-103*, A-to-C; and *cat5-104*, A-to-T). Quantification results of (D) and (F) were summarized as bar graphs in (E) and (G), respectively. The cells were cultured at 30˚C in YPD (D and E) or in YPGly (F and G) media. Relative abundance of Cat5p, normalized with that of Srp1p, in n ≥ 3 in (D) and n ≥ 4 in (F). The mean value for the WT was set to 1.0 (Student's *t*-test, **, $p < 0.01$; *, $p < 0.05$; ns, not significant).

the non-canonical binding sequence UGUAUAAA, rather than only a lack of Puf3p binding. A similar Cat5p expression pattern was also observed in the *cat5-101* cells grown in YPGly (Fig 3C, lanes 2 and 4, RA = 0.36±0.10 and 0.33±0.27, respectively; Student's *t*-test, *p* = 0.448). Thus, Puf3p binding to the non-canonical Puf3p-binding site in the *CAT5* 3′-UTR is not required for yeast growth in the respiratory medium, but contributes to the downregulation of Cat5p translation. Moreover, the non-canonical Puf3p-binding site seems to recruit an unidentified factor(s) that increases Cat5p production under both fermentable and respiratory conditions, and facilitates cooperation with Puf3p in the post-transcriptional regulation of *CAT5*.

We next analyzed *in vivo* effects of a point mutation at position 7 of the non-canonical Puf3p-binding site in the *CAT5* 3′-UTR. A-to-G (*cat5-102*) and A-to-C (*cat5-103*) mutations increased Cat5p expression similarly to *PUF3* deletion in yeast grown in the fermentable medium (Fig 3D and 3E), but these effects were abolished in the respiratory medium (Fig 3F and 3G). Although the *cat5-102* mutant showed a statistically significant difference in Cat5p expression from that in the wild-type cells in the respiratory medium (Student's *t*-test, *p* = 0.012), the difference was so small that it is likely to have little physiological impact. The mutation to the canonical sequence (*cat5-104*; A-to-U) did not affect Cat5p expression under either fermentable or respiratory conditions. These results imply that the A at position 7 of the non-canonical sequence in the *CAT5* Puf3p-binding site is equivalent to the U at position 7 of the canonical sequence with respect to the *in vivo* effects of Puf3p under the conditions analyzed. So far, the reason why the *CAT5* gene contains an A instead of a U at position 7 of its Puf3p-binding site remains to be determined.

We next analyzed the contributions of other Pumillio family proteins to the regulation of *CAT5*. There are 6 *PUF* genes on the yeast genome [21]. Puf4p and Puf5p partially share target mRNAs with Puf3p [23, 72], and Porter *et al*. reported that Puf2p also binds to the *CAT5* mRNA [73]. The *puf4Δ* mutation increased Cat5p expression less than the *puf3Δ* mutation under the fermentable conditions, suggesting that Puf3p makes a major, and Puf4p makes a minor, contribution to the downregulation of Cat5p (Fig 4A and 4B). Interestingly, *puf6Δ* had a small but reproducible opposing effect on Cat5p expression (Fig 4A and 4B). This result suggests that Puf6p may counteract the effects of Puf3p (and Puf4p) and that this effect is abolished by the *cat5-101* mutation. However, statistical analyses showed a difference in the Cat5p expression of the *puf6Δ* and *cat5-101* strains (Student's *t*-test, *p* = 0.027; one-way ANOVA with WT strain, *p* = 0.0014). Therefore, a lack of access of Puf6p to the *CAT5* mRNA fails to completely explain the downregulation of Cat5p by the *cat5-101* mutation. Under the respiratory conditions, *PUF3* deletion, but not *PUF4* or *PUF6* deletion, altered Cat5p expression (Fig 4C and 4D), indicating that only Puf3p regulates Cat5p expression, irrespective of the prevalent type of carbon metabolism. The *cat5-101* mutation reduced Cat5p production in the respiratory medium to a similar level to that obtained under fermentable conditions, which suggests that a factor that binds to this region other than Puf6p plays a major role in the upregulation of Cat5p expression.

## Cat5p protein level is mainly determined by mRNA level: *Puf3Δ* mutation affects *CAT5* mRNA stability while *cat5-101* mutation seems to alter its transcription

To understand the regulatory mechanism of Cat5p level by Puf3p and the non-canonical Puf3p-binding site more in detail, we asked whether the higher Cat5p expression in the *puf3Δ* mutant is the result of an alteration in production and/or stability of protein and/or mRNA. First, we performed a cycloheximide (CHX) chase in the wild-type, *cat5-101*, and *puf3Δ* strains

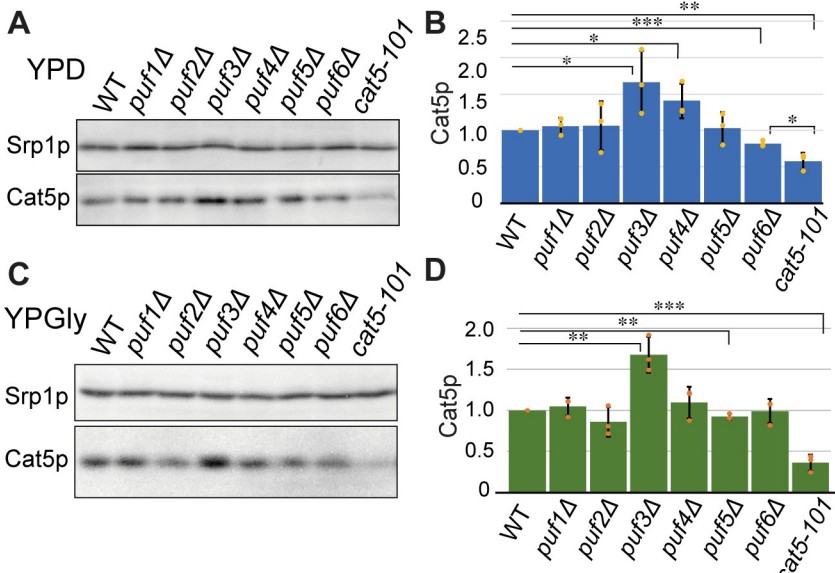

**Fig 4. Effects of deletions in various PUF gene family members on Cat5p expression.** (A) Western blot analysis of Cat5p expression in the wild-type (WT) and the *puf1Δ–puf6Δ* strains. Cells were cultured at 30°C in YPD, and total cell extracts were analyzed as in Fig 3D. (B) Relative abundance of Cat5p, normalized with that of Srp1p, in n ≥ 3 in (A). The mean values for the WT were set to 1.0 (Student's *t*-test, *, $p < 0.05$; **, $p < 0.01$; ***, $p < 0.001$). (C) Western blot analysis of Cat5p expression in the WT and the *puf1Δpuf6Δ* strains grown at 30°C in YPGly. (D) Relative abundance of Cat5p, normalized with that of Srp1p, in n ≥ 3 in (C). The mean value for the WT was set to 1.0 (Student's *t*-test, **, $p < 0.01$; ***, $p < 0.001$).

to determine whether the changes were the results of an alteration in translation or degradation. The *puf3Δ* mutant consistently possessed more Cat5p than the wild-type strain over 0–240 min following the addition of CHX (Fig 5A). The calculated half-lives of Cat5p in the wild-type, *cat5-101*, and *puf3Δ* strains were 4.7±1.2 hr, 6.7±3.3 hr, and 3.8±1.2 hr, respectively, and there were no significant differences among these, according to Student's *t*-test and one-way ANOVA. Then, we examined Cat5p synthesis by L-homopropargylglycine (HPG) pulse-label experiments (Fig 5B). Relative abundance of HPG-labeled Cat5p compared to that of the wild-type cells are 0.65±0.17 in the *cat5-101* and 1.65±0.27 in *puf3Δ* mutants (Student's *t*-test, $p = 0.0125$ for the *cat5-101* and $p = 0.0071$ for *puf3Δ*). Finally, the half-lives of *CAT5* mRNA were measured by a thiolutin chase assay, and found that the *puf3Δ* mutation was associated with a near doubling of the half-life (20±2 min) *versus* the wild-type strain (Fig 5C, 9.8±5.7 min; Student's *t*-test, $p = 0.025$), according to previously reported data regarding mRNA destabilization by Puf3p [36, 37, 48]. Further interestingly, the *cat5-101* mutant expressed less *CAT5* mRNA (Fig 5C), consistent with the Cat5p expression (Fig 3B and 3C), but the half-life of the mRNA (8.8±3.8 min) was comparable to that of the wild-type strain. Thus, the lower *CAT5* mRNA expression in the *cat5-101* strain was not solely the result of a post-transcriptional defect but was also caused by a transcriptional defect. These results indicate that Cat5p levels are controlled by production but not by degradation of Cat5p, and that mRNA stability/transcription affects Cat5p production in addition to translation efficiency.

## Discussion

Recent multi-omics analyses have revealed that yeast Puf3p directly regulates CoQ biosynthesis, in particular *via* Coq5p [74], which catalyzes the immediately prior reaction step of the one

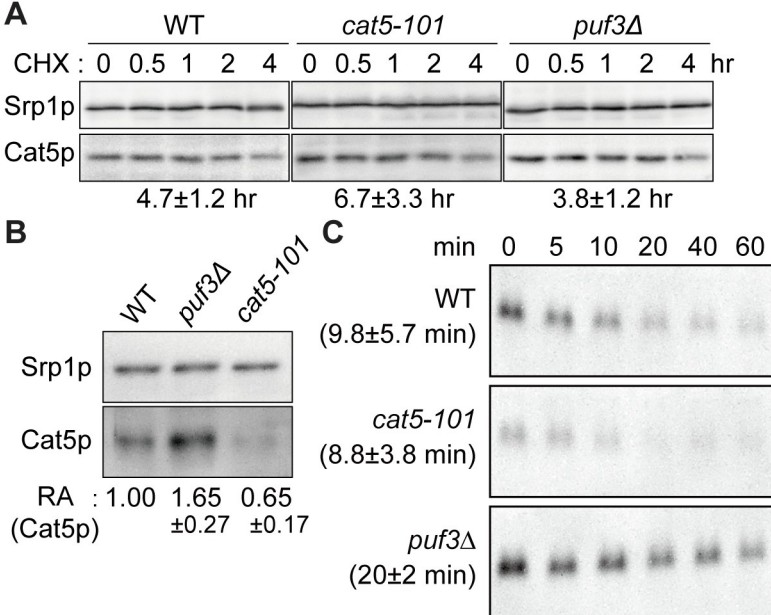

**Fig 5. Stabilization of Cat5p, demonstrated using a CHX chase, and the half-lives of *CAT5* mRNAs, estimated using a thiolutin chase.** (A) Cycloheximide (CHX) chase in the wild-type (WT), *cat5-101*, and *puf3Δ* strains. The Cat5p expression at the time points indicated was determined after the inhibition of protein synthesis by the addition of 200 μg/ml CHX to log-phase cells grown in YPD. Cell lysates were subjected to SDS-PAGE, followed by immunodetection with anti-Cat5p antibodies. Owing to the low signal intensity for Cat5p in the *cat5-101* mutant, the amount of sample loaded was increased 2.25-fold. Calculated half-lives (means of $n \geq 3$ independent experiments) are shown in below of the images, along with standard deviations. (B) The HPG pulse-labelling in the WT, *puf3Δ*, and *cat5-101* strains. Logarithmically grown cells in the SD-based medium were treated with methionine starvation for 30 min, and labeled with HPG for 30 min. HPG-labeled proteins were biotinylated by click chemistry and purified with avidin D-agarose. The numbers under the gel images represent the mean ± standard deviation of the relative amount (RA) of HPG-labeled Cat5p, quantified from $n \geq 3$. The WT Cat5p was set to 1.00. (C) Thiolutin chase in the WT, *cat5-101*, and *puf3Δ* strains. Cells were cultured in YPD to the log phase, then thiolutin was added to stop transcription. Samples were collected at the indicated time points. RNA prepared from the samples (5.0 μg RNA/lane) was separated on a 2.2 M formaldehyde/1.2% agarose gel and subjected to northern blotting using a probe against the *CAT5* ORF. Signals were quantified and fitted to an exponential curve to obtain a *CAT5* mRNA half-life for each strain. A representative northern blot for each strain is shown. The calculated half-lives (means of 3 or 4 independent experiments) are shown in parentheses, along with standard deviations.

*via* Cat5p [56]. Indeed, yeast strains lacking Puf3p are deficient in CoQ synthesis under fermentable conditions but not under respiratory conditions [74]. Although HITS-CLIP and RNA tagging data led to the eventual exclusion of *CAT5* from the list of highly likely Puf3p targets [74], we identified *CAT5* as another regulatory point of CoQ biosynthesis by Puf3p, and this regulation may be utilized to fine-tune CoQ biosynthesis. In the present study, we have shown that Puf3p directly interacts with a non-canonical Puf3p-binding sequences containing a variation at position 7 in the *CAT5* and *MRPL16* 3'-UTRs *in vitro* (Fig 2). Our *in vitro* analysis of these 3'-UTRs shows that Puf3p accepts U-to-C and U-to-A variants at position 7 of the canonical 8-nt binding sequence. Thus, it is possible that Puf3p recognizes a wider range of mRNAs *in vivo* than just the strict targets identified in previous studies [25, 26, 74]. In addition, some mRNA species may have multiple canonical and/or non-canonical Puf3p-binding sites with redundant functions. Of note, in *in vivo* mutant analysis of *CAT5*, the Puf3p effect on Cat5p expression seemed to be abolished by introduction of C at position 7 in the non-canoical site while Puf3p bound to *MRPL16* 3'-UTR with the U-to-C variant *in vitro*. Flanking sequences of the Puf3p-binding site may affect Puf3p affinity to the 3-UTR with position 7

variants. Or as discussed later, seuqnece preference of another factor(s) recognizing the *CAT5* 3'-UTR region near the non-canonical Puf3p-binding site may also alter the effect of mutation at the position 7.

The N-terminal part of the PUF domain, responsible for recognizing the 3′ region of the Puf-binding site, displays greater flexibility in terms of its acceptance of target-nucleotide mutations than the C-terminal part [75–77]. Indeed, analysis of human PUM2 binding sites using SEQRS *in vitro* and PAR-CLIP *in vivo* revealed that substitutions to A or C at position 7 of the canonical sequence occur naturally [75, 78]. In addition, several PUF proteins show broader specificity when certain undesirable nucleotides are absent [34, 79]. We do not know how Puf3p-RD structurally can recognize these variant sequences, but as proposed by Zhou *et al.* [76, 77], an equilibrium between the individual binding specificity of each repeat to the corresponding nucleotide and the total binding affinity for target mRNAs in PUF proteins may be crucial.

The mutation studies revealed that Cat5p expression is regulated not only by Puf3p but also by other proteins, including Puf4p and Puf6p, *in vivo* (Fig 4). Puf3p more potently reduces *CAT5* expression in fermentable medium than in respiratory medium, whereas the other Puf proteins only have effects under fermentable conditions. *CAT5* mRNA expression was not increased, but was in fact reduced, by the *cat5-101* mutation (UGUA to ACAC) in the non-canonical Puf3p-binding site, which should have mimicked *PUF3* deletion (Fig 5C). The *cat5-101* mutation led to a ~70% reduction in Cat5p production in both the fermentable and respiratory media (Fig 3B and 3C). Thiolutin chase analysis revealed that the *cat5-101* mutant reduced the abundance but not the stability of the *CAT5* mRNA (Fig 5C). Therefore, it appears that the non-canonical Puf3p-binding sequence is also necessary for the correct transcription of *CAT5* mRNA. Moreover, the results suggest that the non-canonical Puf3p-binding site is recognized not only by factors that destabilize *CAT5* mRNA, such as Puf3p, but also by one or more factors that stabilize it. The abolition of the interaction of both destabilizing and stabilizing factors may cause this wild-type level stability of the *cat5-101* mRNA. Puf6p is not the only candidate for this stabilizer. According to a study of diverse sequence motifs that are enriched in mRNAs bound by specific RNA-binding proteins [24], Pab1p has the ability to bind the non-canonical Puf3p-binding sequence UGUAUAAA, and therefore may be a candidate. Pab1p generally controls poly(A) tail length and promotes efficient translation through binding to poly(A) tails, but it also binds to the 3′-UTRs of certain mRNAs and regulates their fates [80–82]. The analysis of point mutations at position 7 of the non-canonical Puf3p-binding site in the *CAT5* 3′-UTR revealed that the A-to-G and A-to-C mutations partially mimic the *puf3Δ* phenotype under fermentable conditions, whereas Puf3p reduces Cat5p, but not Mrpl16p, production, even in respiratory medium (Figs 1 and 3D–3G). This may imply that the non-canonical Puf3p-binding site is recognized by Puf3p only under fermentable conditions, as for the *MRPL16* mRNA, and that the higher Cat5p expression in *puf3Δ* cells grown in glycerol-containing medium is the result of a secondary effect of *PUF3* deletion. It is also possible that the modification of Puf3p, such as phosphorylation, under respiratory conditions may alter the Puf3p-binding preference at position 7 of this non-canonical binding site.

Comprehensive analyses are required to fully understand translational regulation by Puf3p. A number of questions remain to be answered. Firstly, how wide is the range of non-canonical RNA sequences recognized by Puf3p? Secondly, how does Puf3p reduce Cat5p expression under both fermentable and respiratory conditions, while Puf3p affects Mrpl16p expression only under fermentable conditions? Finally, what are the unknown factor(s) and their specific functions? To address these questions and comprehend Puf3p's involvement in Cat5p regulation, additional biochemical analysis is imperative. This includes evaluating $CoQ_6$ levels in *cat5* mutants with disrupted Puf3p binding. Furthermore, isolating mutants deficient in the

downregulation of Cat5p would be valuable to identify the *cis*-element for the potential negative regulator.

Importantly, the fact that hCOQ7 complements yeast lacking Cat5p highlights its cross-species functionality and the utility of yeast as a model for investigating hCOQ7 mutations associated with $CoQ_{10}$ deficiency and related diseases. Our research findings have the potential to bridge the divide between yeast and human CoQ biology, offering insights into precise mechanism of *hCOQ7* regulation at the mRNA level in response to expression of other hCOQ proteins and CoQ metabolic states, and leading to potential therapeutic approaches for CoQ deficiency-related disorders. The enhancing our comprehension of the Puf3p-driven regulation in yeast would also contribute to understanding the pathogenesis of CoQ deficiency in humans.

## Supporting information

**S1 Fig. Loss of Puf3p modestly increases the mRNA expression of *RSM10*, *MRPL16*, and *CAT5*, but reduces that of *AIM17*.** (A) Northern blot analysis of several monosome-enriched mRNAs. Total RNAs were isolated from the wild-type (W) and *puf3*Δ (Δ) yeasts grown in the fermentable medium (YPD), and those were subjected to the northern blotting for the indicated mRNA species. The numbers represent relative expression changes of mRNA abundance. The means and standard deviations of the mRNA signals (Δ/W, *puf3*Δ/WT) were calculated from n ≥ 3. Lower panels represent the total RNAs in each corresponding sample, visualized by GelRed staining prior to northern blotting. (B) Northern blot analysis of the *MRPL16* and *CAT5* mRNAs extracted from yeasts grown in respiratory media (YPGal and YPGly). The relative expression changes of the mRNAs were analyzed as described above. Lower panels represent the total RNAs in each corresponding sample, visualized by GelRed staining prior to northern blotting.
(PDF)

**S2 Fig. Higher temperature (37˚C) did not enhance the phenotypes in yeasts harboring the *cat5-101* mutation.** (A) Growth comparison among wild-type, *cat5-101*, and *cat5*Δ strains in the presence or absence of the *PUF3* gene. Saturated cultures of the indicated strains were serially diluted by 10-fold as shown in the bottom and dropped onto YPD or YPGly plates, and incubated at 37˚C. (B) Western blot analysis of Cat5p. The yeast strains in (A) were grown at 37˚C in YPD, and subjected to western blotting. Srp1p was used as a loading control. The numbers under the gel images represent the mean ± standard deviation of the relative amount (RA) of Cat5p, quantified from n ≥ 3. The WT Cat5p expression was set to 1.00.
(PDF)

**S1 Table. Yeast Strains used in this study.**
(XLSX)

**S2 Table. Plasmids used in this study.**
(XLSX)

**S3 Table. Primers used in this study.**
(XLSX)

**S1 Raw images.**
(PDF)

**S2 Raw images.**
(PDF)

**S3 Raw images.**
(PDF)

**S4 Raw images.**
(PDF)

**S5 Raw images.**
(PDF)

**S6 Raw images.**
(PDF)

## Acknowledgments

We are grateful to Prof. Antonio Barrientos (University of Miami, USA) and Prof. Catherine F. Clarke (UCLA, USA) for providing the α-Mrpl16p and α-Cat5p/Coq7p antibodies. We also thank Dr. Shintaro Iwasaki (RIKEN, Japan) for technical advices and discussion.

## Author Contributions

**Conceptualization:** Sachiko Hayashi, Tohru Yoshihisa.

**Data curation:** Sachiko Hayashi, Tohru Yoshihisa.

**Formal analysis:** Sachiko Hayashi, Tohru Yoshihisa.

**Funding acquisition:** Sachiko Hayashi, Tohru Yoshihisa.

**Investigation:** Sachiko Hayashi, Kazumi Iwamoto, Tohru Yoshihisa.

**Project administration:** Sachiko Hayashi.

**Resources:** Sachiko Hayashi, Tohru Yoshihisa.

**Supervision:** Sachiko Hayashi, Tohru Yoshihisa.

**Validation:** Sachiko Hayashi, Kazumi Iwamoto, Tohru Yoshihisa.

**Visualization:** Sachiko Hayashi, Kazumi Iwamoto, Tohru Yoshihisa.

**Writing – original draft:** Sachiko Hayashi, Tohru Yoshihisa.

**Writing – review & editing:** Sachiko Hayashi, Tohru Yoshihisa.

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
