## [Decision Letter · Decision Letter 0]

5 Oct 2023

PONE-D-23-24795A non-canonical Puf3p-binding sequence regulates CAT5/COQ7 mRNA under both fermentable and respiratory conditions in budding yeast.PLOS ONE

Dear Dr. Hayashi,

Thank you for submitting your manuscript to PLOS ONE. After careful consideration, we feel that it has merit but does not fully meet PLOS ONE’s publication criteria as it currently stands. Therefore, we invite you to submit a revised version of the manuscript that addresses the points raised during the review process.

We look forward to receiving your revised manuscript.

Kind regards,

Patrick Lajoie, PhD

Academic Editor

PLOS ONE

Journal Requirements:

Additional Editor Comments:

As you can see below, the revisions are not extensive and should be easy to address in the text. 

Reviewers' comments:

Reviewer's Responses to Questions

**Comments to the Author**

1. Is the manuscript technically sound, and do the data support the conclusions?

Reviewer #1: Yes

Reviewer #2: Yes

2. Has the statistical analysis been performed appropriately and rigorously? 

Reviewer #1: Yes

Reviewer #2: Yes

3. Have the authors made all data underlying the findings in their manuscript fully available?

Reviewer #1: Yes

Reviewer #2: Yes

4. Is the manuscript presented in an intelligible fashion and written in standard English?

Reviewer #1: Yes

Reviewer #2: Yes

5. Review Comments to the Author

Reviewer #1: In this study, the authors have systematically demonstrated that Puf3p directly targets CAT5 mRNA via its non-canonical Puf3p binding sequence and that Puf3p represses translation of Cat5p. The authors also show that related PUF proteins do not share the same potency as regulators Cat5p expression. Overall, the findings present a novel regulatory mechanism of Puf3p on Cat5p expression and a broader range of recognition of Puf3p mRNA targets in different cell-growth conditions than was previously understood.

Minor comments:

Abstract, line 16: Add “The” before Saccharomyces cerevisiae. The authors may consider simplifying the sentence or breaking in into separate points to improve clarity.

Line 22-23: It is unclear to this reviewer what “…Puf3p has global pleiotropic effects of Puf3p on gene expression under fermentable conditions.” Refers to. This reviewer suggests changing “of Puf3p” to “on Puf3p targets” if this is accurate. Or the removal of “of Puf3p”.

General comments:

Conclusion, line 417: The authors conclude with a statement relating to the translational application of their results to CoQ deficiency in humans. This reviewer agrees with this statement and the potential impact of these findings. However, introducing this connection at an earlier point in the manuscript (such as the Abstract and/or Introduction) and elaborating on what is not well understood about CAT5 regulation in humans may increase the accessibility and impact of the presented findings beyond yeast.

Further, the relevance of Puf3p to this statement is not readily clear to this reviewer as the human orthologs appear to regulate different functional targets. This reviewer suggests breaking the concluding remark into separate sentences for clarity.

Reviewer #2: In this study, the authors shown that Puf3p, bind to the mRNA of CTA5 via a non-canonical Puf3 binding sequence. Cat5 with modified binding do not have a growth defect but displayed altered CAT5 expression. This is specific to Puf3 as other other PUF family members do not show the same phenotype. Data suggest that the Puf3 binding set may serve in recruiting other factors to control CAT% abundance.

Overall, I feel the data are convincing and support the the conclusions which is in agreement with the PLoS One requirements. I have only minor comments:

1-Abstract: "and that Puf3p has global pleiotropic effects of Puf3p on gene expression under fermentable conditions." please rephrase

2-I think a Figure summarizing the presence of Puf3 binding motifs in its absence in CAT5 and the localization of the non-canonical sequence with add clarity to the manuscript instead having to refer to supplemental figures.

3-Bar graphs should include individual data points

4-This reviewers would have love to see some ideas of potential additional regulators in the discussion.

6. PLOS authors have the option to publish the peer review history of their article (what does this mean?). If published, this will include your full peer review and any attached files.

Reviewer #1: No

Reviewer #2: No

---

## [Author Response · Author response to Decision Letter 0]

19 Nov 2023

Journal Requirements:

Thank you for your guidance. We have reviewed PLOS ONE's style requirements. Our revised manuscript adheres to the recommended style templates. If you have any additional recommendations or specific aspects you would like us to revisit, please let us know.

Thank you for your guidance. We provided the original images corresponding to all blot/gel results reported in the main and supporting figures in Supporting Information as pdf files named "Raw_images X.pdf". 

We found one reference shown below had been corrected by the Publisher. Therefore, we added a citation and a full reference for the correction notice.

76. Zhou W, Melamed D, Banyai G, Meyer C, Tuschl T, Wickens M, et al. Expanding the binding specificity for RNA recognition by a PUF domain. Nat Commun. 2021;12: 5107. doi:10.1038/s41467-021-25433-6.

In addition, we added nine references due to the revised description of the main text. Added references are below.

54. Padilla S, Tran UC, Jiménez-Hidalgo M, López-Martín JM, Martín-Montalvo A, Clarke CF, et al. Hydroxylation of demethoxy-Q6 constitutes a control point in yeast coenzyme Q6 biosynthesis. Cell Mol Life Sci. 2009;66: 173–186. doi:10.1007/s00018-008-8547-7.

55. Vajo Z, King LM, Jonassen T, Wilkin DJ, Ho N, Munnich A, et al. Conservation of the Caenorhabditis elegans timing gene clk-1 from yeast to human: a gene required for ubiquinone biosynthesis with potential implications for aging. Mamm Genome. 1999;10: 1000–1004. doi:10.1007/s003359901147.

57. Freyer C, Stranneheim H, Naess K, Mourier A, Felser A, Maffezzini C, et al. Rescue of primary ubiquinone deficiency due to a novel COQ7 defect using 2,4-dihydroxybensoic acid. J Med Genet. 2015;52: 779–783. doi:10.1136/jmedgenet-2015-102986.

58. Herebian D, Seibt A, Smits SHJ, Bünning G, Freyer C, Prokisch H, et al. Detection of 6-demethoxyubiquinone in CoQ10 deficiency disorders: insights into enzyme interactions and identification of potential therapeutics. Mol Genet Metab. 2017;121: 216–223. doi:10.1016/j.ymgme.2017.05.012.

59. Cascajo M V., Abdelmohsen K, Noh JH, Fernández-Ayala DJM, Willers IM, Brea G, et al. RNA-binding proteins regulate cell respiration and coenzyme Q biosynthesis by post-transcriptional regulation of COQ7. RNA Biol. 2016;13: 622–634. doi:10.1080/15476286.2015.1119366.

60. Bohn JA, Van Etten JL, Schagat TL, Bowman BM, McEachin RC, Freddolino PL, et al. Identification of diverse target RNAs that are functionally regulated by human Pumilio proteins. Nucleic Acids Res. 2018;46: 362–386. doi:10.1093/nar/gkx1120.

61. Quinzii CM, Emmanuele V, Hirano M. Clinical presentations of coenzyme Q10 deficiency syndrome. Mol Syndromol. 2014;5: 141–146. doi:10.1159/000360490.

62. Wongkittichote P, Lasio MLD, Magistrati M, Pathak S, Sample B, Carvalho DR, et al. Phenotypic, molecular, and functional characterization of COQ7-related primary CoQ10 deficiency: Hypomorphic variants and two distinct disease entities. Mol Genet Metab. 2023;139: 107630. doi:10.1016/j.ymgme.2023.107630.

63. Alcázar-Fabra M, Rodríguez-Sánchez F, Trevisson E, Brea-Calvo G. Primary Coenzyme Q deficiencies: A literature review and online platform of clinical features to uncover genotype-phenotype correlations. Free Radic Biol Med. 2021;167: 141–180. doi:10.1016/j.freeradbiomed.2021.02.046.

Additional Editor Comments:

Point-to-point answers

5. Review Comments to the Author

Reviewer #1: In this study, the authors have systematically demonstrated that Puf3p directly targets CAT5 mRNA via its non-canonical Puf3p binding sequence and that Puf3p represses translation of Cat5p. The authors also show that related PUF proteins do not share the same potency as regulators Cat5p expression. Overall, the findings present a novel regulatory mechanism of Puf3p on Cat5p expression and a broader range of recognition of Puf3p mRNA targets in different cell-growth conditions than was previously understood.

Minor comments:

Abstract, line 16: Add “The” before Saccharomyces cerevisiae. The authors may consider simplifying the sentence or breaking in into separate points to improve clarity.

Thank you for your suggestion. We rephrased the corresponding part as “The Saccharomyces cerevisiae” on p. 2, line 22.

Line 22-23: It is unclear to this reviewer what “…Puf3p has global pleiotropic effects of Puf3p on gene expression under fermentable conditions.” Refers to. This reviewer suggests changing “of Puf3p” to “on Puf3p targets” if this is accurate. Or the removal of “of Puf3p”

Thank you for your advice. We revised the sentence, including the corresponding part as below.

p.2, lines 26–28:

Puf3p preferentially binds to 8-nt conserved binding sequences in the 3′-UTR of nuclear-encoded mitochondrial (nc-mitochondrial) mRNAs, leading to broad effects on gene expression under fermentable conditions.

General comments:

Conclusion, line 417: The authors conclude with a statement relating to the translational application of their results to CoQ deficiency in humans. This reviewer agrees with this statement and the potential impact of these findings. However, introducing this connection at an earlier point in the manuscript (such as the Abstract and/or Introduction) and elaborating on what is not well understood about CAT5 regulation in humans may increase the accessibility and impact of the presented findings beyond yeast.

We appreciate your valuable comment and agree with your point. We added the descriptions in the abstract, introduction, and discussion below.

Abstract, p. 2, lines 40–42:

Given that pathological variants of human COQ7 lead to CoQ10 deficiency and yeast cat5∆ can be complemented by hCOQ7, our findings may also offer some insights into clinical aspects of COQ7-related disorders.

Introduction, p. 5, lines 100–117:

Yeast CAT5/COQ7 mRNA encodes a putative monooxygenase required for coenzyme Q (CoQ) biosynthesis and its product Cat5p is an integral membrane protein in the inner mitochondrial membrane [51–53]. Cat5p expression is regulated at the level of mRNA, especially in response to carbon source and CoQ-related metabolites though precise molecular mechanism of this regulation is still under investigation [51,54]. The functional conservation of Cat5p/Coq7p among species is shown by the ability of human COQ7 (hCOQ7) to rescue yeast CoQ6 deficiency caused by cat5Δ [55,56]. Indeed, expression/stability of hCOQ7 is fine-tuned by the level of hCOQ4, and is also affected by 2,4-dihydroxybenzoic acid, which is capable of bypassing the enzymatic step performed by hCOQ7 [57,58]. Such fine-tuning may be achieved at the mRNA level, like the case of yeast Cat5p. Various RNA-binding proteins like HuR and hnRNP C1/C2 interact with the 3′-UTR of hCOQ7 mRNA to modulate hCOQ7 levels, thereby controlling CoQ10 [59]. The human Pumilio proteins (PUMs), PUM1 and PUM2, also bind to the 3′-UTR of hCOQ7 mRNA via their canonical binding motif, while the expression level of hCOQ7 mRNA is unchanged in the PUM-knockdown cells [60]. Mutations in hCOQ7 are associated with primary ubiquinone deficiency, which contributes to CoQ10 deficiency syndrome, and related diseases, predominantly featuring hypertonia and sensorineural hearing loss (SNHL) [57,61–63]. To fully understand the pathogenesis of CoQ10-deficiency related diseases, not only enzymatic mechanism of COQ proteins but also regulation of their expression needs to be clarified.

Discussion, p.26, lines 612–616:

Importantly, the fact that hCOQ7 complements yeast lacking Cat5p highlights its cross-species functionality and the utility of yeast as a model for investigating hCOQ7 mutations associated with CoQ10 deficiency and related diseases. Our research findings have the potential to bridge the divide between yeast and human CoQ biology, offering insights into precise mechanism of hCOQ7 regulation at the mRNA level in response to expression of other hCOQ proteins and CoQ metabolic states, and leading to potential therapeutic approaches for CoQ deficiency-related disorders. The enhancing our comprehension of the Puf3p-driven regulation in yeast would also contribute to understanding the pathogenesis of CoQ deficiency in humans.

Further, the relevance of Puf3p to this statement is not readily clear to this reviewer as the human orthologs appear to regulate different functional targets. This reviewer suggests breaking the concluding remark into separate sentences for clarity.

Thank you for your thoughtful advice. We revised the corresponding part as below.

p. 25 and 26, lines 604–616:

To address these questions and comprehend Puf3p's involvement in Cat5p regulation, additional biochemical analysis is imperative. This includes evaluating CoQ6 levels in cat5 mutants with disrupted Puf3p binding. Furthermore, isolating mutants deficient in the downregulation of Cat5p would be valuable to identify the cis-element for the potential negative regulator.

Importantly, the fact that hCOQ7 complements yeast lacking Cat5p highlights its cross-species functionality and the utility of yeast as a model for investigating hCOQ7 mutations associated with CoQ10 deficiency and related diseases. Our research findings have the potential to bridge the divide between yeast and human CoQ biology, offering insights into precise mechanism of hCOQ7 regulation at the mRNA level in response to expression of other hCOQ proteins and CoQ metabolic states, and leading to potential therapeutic approaches for CoQ deficiency-related disorders. The enhancing our comprehension of the Puf3p-driven regulation in yeast would also contribute to understanding the pathogenesis of CoQ deficiency in humans.

Reviewer #2: In this study, the authors shown that Puf3p, bind to the mRNA of CTA5 via a non-canonical Puf3 binding sequence. Cat5 with modified binding do not have a growth defect but displayed altered CAT5 expression. This is specific to Puf3 as other other PUF family members do not show the same phenotype. Data suggest that the Puf3 binding set may serve in recruiting other factors to control CAT% abundance.

Overall, I feel the data are convincing and support the the conclusions which is in agreement with the PLoS One requirements. I have only minor comments:

1-Abstract: "and that Puf3p has global pleiotropic effects of Puf3p on gene expression under fermentable conditions." please rephrase

We apologize for the improper expression in the previous manuscript. We rephrased it to “leading to broad effects on gene expression under fermentable conditions” on p. 2, lines 27 and 28.

2-I think a Figure summarizing the presence of Puf3 binding motifs in its absence in CAT5 and the localization of the non-canonical sequence with add clarity to the manuscript instead having to refer to supplemental figures.

Thank you for your suggestion. We incorporated it by moving Supplemental Table 1 to the main text and designating it as Table 1.

3-Bar graphs should include individual data points

Thank you for your comment. We revised the bar graphs to include individual data points. Please see Figs. 1, 3, and 4.

4-This reviewers would have love to see some ideas of potential additional regulators in the discussion.

We sincerely appreciate your suggestion to include ideas for potential additional regulators in the discussion. We aim to express ideas with confidence only when supported by robust evidence. Currently, we unfortunately lack solid data detailing regulatory mechanisms for CAT5 expression via Puf3p and others. While we refrain from delving into speculative regulators in the present manuscript, we are moving forward with research to explore and validate additional regulatory elements. We only slightly add the description on p.25 and 26, lines 604–608: 

To address these questions and comprehend Puf3p's involvement in Cat5p regulation, additional biochemical analysis is imperative. This includes evaluating CoQ6 levels in cat5 mutants with disrupted Puf3p binding. Furthermore, isolating mutants deficient in the downregulation of Cat5p would be valuable to identify the cis-element for the potential negative regulator. 

Thank you again for your thoughtful comments.

---

## [Editor Report · Decision Letter 1]

24 Nov 2023

A non-canonical Puf3p-binding sequence regulates CAT5/COQ7 mRNA under both fermentable and respiratory conditions in budding yeast.

PONE-D-23-24795R1

Dear Dr. Hayashi,

We’re pleased to inform you that your manuscript has been judged scientifically suitable for publication and will be formally accepted for publication once it meets all outstanding technical requirements.

Kind regards,

Patrick Lajoie, PhD

Academic Editor

PLOS ONE

---

## [Editor Report · Acceptance letter]

7 Dec 2023

PONE-D-23-24795R1 

A non-canonical Puf3p-binding sequence regulates *CAT5/COQ7* mRNA under both fermentable and respiratory conditions in budding yeast. 

Dear Dr. Hayashi:

I'm pleased to inform you that your manuscript has been deemed suitable for publication in PLOS ONE. Congratulations! Your manuscript is now with our production department. 

Kind regards, 

on behalf of

Dr. Patrick Lajoie 

Academic Editor

PLOS ONE